# Hoppity: Learning Graph Transformations to Detect and Fix Bugs in Programs

**Elizabeth Dinella**[*]
University of Pennsylvania

**Hanjun Dai**[*]
Google Brain

**Ziyang Li**
University of Pennsylvania

**Mayur Naik**
University of Pennsylvania

**Le Song**
Georgia Tech

**Ke Wang**
Visa Research

## Abstract

We present a learning-based approach to detect and fix a broad range of bugs in Javascript programs. We frame the problem in terms of learning a sequence of graph transformations: given a buggy program modeled by a graph structure, our model makes a sequence of predictions including the position of bug nodes and corresponding graph edits to produce a fix. Unlike previous works built upon deep neural networks, our approach targets bugs that are more diverse and complex in nature (i.e. bugs that require adding or deleting statements to fix). We have realized our approach in a tool called Hoppity. By training on 290,715 Javascript code change commits on Github, Hoppity correctly detects and fixes bugs in 9,490 out of 36,361 programs in an end-to-end fashion. Given the bug location and type of the fix, Hoppity also outperforms the baseline approach by a wide margin.

## 1 Introduction

The sheer size and complexity of modern codebases makes it impossible for them to be bug-free. As a result, a more reasonable and effective strategy has emerged, which aims to prevent bugs in production by applying automated tools to detect and even fix them early in the development process.

This trend has gained increasing popularity in recent years. Examples include Google's Tricorder (Sadowski et al., 2015), Facebook's Getafix (Scott et al., 2019) and Zoncolan, and Microsoft's Visual Studio IntelliCode. The techniques underlying these tools can be classified into broadly two categories: logical, rule-based techniques (Sadowski et al., 2015) and statistical, data-driven techniques (Allamanis et al., 2018; Pradel & Sen, 2018; Vasic et al., 2019). The former uses manually written rules capturing undesirable code patterns and scans the entire codebase for these classes of bugs. The latter learns to detect abnormal code from a large code corpus using deep neural networks. Despite great strides, however, both kinds of tools are limited in generality because they target error patterns in specific codebases or they target specific bug types. For instance, Zoncolan's rules are designed to be specifically applicable to Facebook's codebases, and deep learning models target specialized bugs in variable naming (Allamanis et al., 2018) or binary expressions (Pradel & Sen, 2018). Moreover, the patterns are relatively syntactic, allowing them to be specified by human experts using logical rules or learnt from a corpus of programs.

In this paper, we propose a novel learning-based approach for finding and fixing bugs in Javascript programs automatically. Javascript is a scripting language designed for web application development. It has been the most popular programming language on GitHub since 2014 (Github, 2019). Repairing Javascript code presents a unique challenge as bugs manifest in diverse forms due to unusual language features and the lack of tooling support. Therefore, the primary goal of our approach is generality since it must be effective against a board spectrum of programming errors, such as using wrong operators or identifiers, accessing undefined properties, mishandling variable scopes, triggering type incompatibilites, among many others. Another important novel aspect concerns our approach's ability to deal with bugs that are more complex and semantic in nature, namely, bugs that require adding or removing statements from a program, which are not considered by prior works. Finally, compared to automated program repair techniques (Le Goues et al., 2019; Scott et al., 2019; Hua et al., 2018; Chen

---

[*]The first two authors contributed equally to this work.

et al., 2018) which require knowledge of bug location, this paper presents an end-to-end approach including localizing bugs, predicting the types of fixes, and generating patches.

We design our model architecture in a similar vein as a Neural Turing Machine (NTM) (Graves et al., 2014). It consists of an external memory (a Graph Neural Network) for embedding a buggy program and a central controller (an LSTM) that makes a sequence of primitive actions (e.g., predicting type, generating patch, etc.) to perform a fix. The multi-step decision process is implemented by an autoregressive model. Crucially, our model differs from the standard NTM in how the memory is manipulated: apart from the common read and write operations, the controller can also expand or shrink the memory when adding or deleting nodes in the original graph.

We have realized our approach in a tool called HOPPITY. By training on 290,715 Javascript code change commits collected from Github, HOPPITY correctly detects and fixes bugs in 9,490 out of 36,361 programs using a beam size of three.

## 2 MOTIVATING EXAMPLES

Javascript is quite different compared to traditional object-oriented languages (e.g. C++, Java, or C#). In addition to the weak, dynamic typing discipline of scripting languages, Javascript supports many peculiar features that do not exist in other languages. For example, it allows a property (i.e., a field) to be added to or removed from an object at runtime. As another example, Javascript did not support block-level scoping until recently, allowing a variable defined in a block structure such as a `for` loop to be exposed to the entire function in which the loop occurs. While the latest ES6 language standard incorporates block-level scoping, developers have been programming without it for decades, resulting in a large body of legacy code. Finally, Javascript's `eval` function, which interprets and executes a string as a code fragment, is widely regarded as a major source of bugs and vulnerabilities. All of these aspects make programming in Javascript a frustrating and error-prone experience.

```javascript
function clearEmployeeListOnLinkClick(){
  document.querySelector("a").addEventListener("click",
    function(event){
      document.querySelector("ul").InnerHTML = "";
    }
  );
}
```

(a) `InnerHTML` should have been `innerHTML`.

```javascript
if (matches) {
  return {
    episode: Number(matches.groups.episode),
    hosts: matches.groups.hosts.split(/([,&]+|\sand\s)/).
              map(el => S(el).trim().s)
  };
}
```

(b) Highlighted parentheses should have been removed.

```javascript
module.exports = function (grunt) {
  grunt.initConfig({
    execute: {...}, copy: {...}, checktextdomain: {...}
    wp_readme_to_markdown: {...}, makepot: {...}})
  ...
  grunt.registerTask('default',['wp_readme_to_markdown'
    ,'makepot','execute','checktextdomain'])
};
```

(c) `copy` function should have also been included in the highlighted list.

```javascript
export default {
  computed: {
    level () {
      return dictMap.skillLevel[
        parseInt((this.value === 0 ? 1 : this.value)/20)];
    }
  },...
}
```

(d) `parseInt` should have been removed because `===` implies `this.value` is an integer.

Figure 1: Example programs that illustrate limitations of existing approaches inculding both rule-based static analyzers and neural-based bug predictors.

Static analyzers aim to detect common coding errors in Javascript programs by applying logical rule-based reasoning on source code. TAJS (Jensen et al., 2009) and ESLint (Zakas, 2013) are prominent examples. These tools face important challenges to be effective. We present several examples in Figure 1 to illustrate their limitations. Due to the complex nature of client-side web APIs, TAJS and ESLint choose to ignore analyzing built-in libraries for the sake of scalability. As depicted in Figure 1a, when developers mistakenly capitalize the first letter of `innerHTML`, a property of class `Element` in DOM (Document Object Model), both analyzers fail to catch the error. Javascript will then silently allow developers to set the previously non-existent property `InnerHTML` to the empty string. Later, when developers attempt to access the intended property, `innerHTML`, the program will crash and potentially cause a security vulnerability or incur a costly debugging experience. Additionally, static analyzers can never deal with functional bugs (i.e., errors that violate the program specification and yet conform to the coding rules). Figure 1b shows one such example. The goal is to split a string using regular expressions. However, the program incorrectly splits the input `' and '` into `['', ' and ', '']` instead of `['', '']`, which is what the developer intended. Since

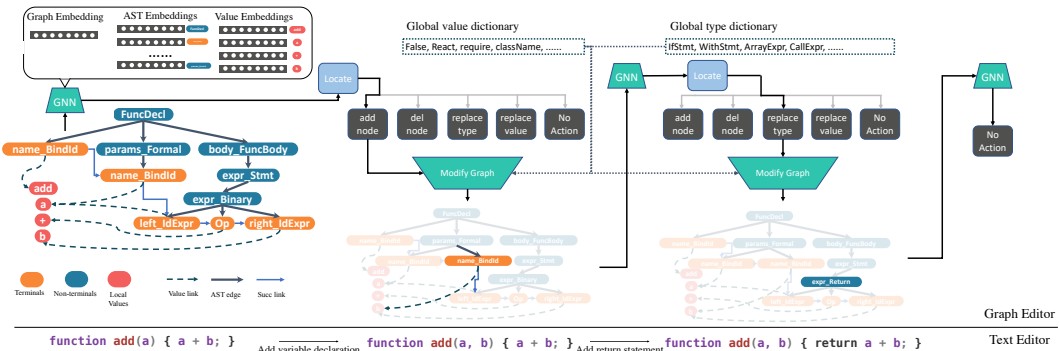

Figure 2: Code repair as graph transformation. Each step the source code graph is edited via one of the operator module until STOP is triggered by controller.

the error is simply a mismatch between the developer's implicit specification and implementation, static analyzers are incapable of catching it.

The bugs that static analyzers missed in both cases are in hindsight quite obvious to human programmers. The criteria they use is very simple: any code snippet that seems to deviate from common code patterns is likely to be buggy. This is precisely the observation that our approach seeks to mimic. In particular, if a model observes a property or an unusual way of splitting strings that never appeared in the training data, it is likely to recognize those abnormal code fragments as potential bugs. The main advantage of our approach over existing neural-based bug detectors (Allamanis et al., 2018; Pradel & Sen, 2018; Vasic et al., 2019) is its generality. Unlike prior works that target specific classes of bugs (e.g., variable naming issues or binary expression bugs), we train a single model to deal with a wide range of bug types, encompassing all previously proposed ones. Compared to past approaches that leverage a graph-based neural network model (Allamanis et al., 2018), our model is capable of more sophisticated transformations such as adding or removing nodes, as shown in Figure 1c and 1d. Finally, our model not only locates but also fixes bugs, whereas program repair (Le Goues et al., 2019; Scott et al., 2019; Hua et al., 2018; Chen et al., 2018) or bug localization (Ball et al., 2003; Jose & Majumdar, 2011; Wang et al., 2019) techniques only solve a single task.

## 3 MODEL

We model the problem of detecting and repairing bugs in programs as a structured prediction problem on a graph-based representation of programs. Given a graph $g_{bug}$ that represents a buggy program, we wish to predict a graph $g_{fix}$ that represents the fixed program. Our model aims to capture the structured prediction by a sequence of up to $T$ steps of graph transformations:

$$p(g_{fix}|g_{bug};\theta) = p(g_1|g_{bug};\theta)p(g_2|g_1;\theta)\ldots p(g_{fix}|g_{T-1};\theta) \tag{1}$$

The high-level overview of the graph sequence transformation is shown in Figure 2. Different programs may need a different number of steps $T(g_{bug})$ which is also determined by the model.

We first introduce our representation module for programs in Sec 3.1. We then elucidate each step of the above transformation in Sec 3.2. Finally, we summarize and present the full model in Sec 3.3.

### 3.1 PROGRAM REPRESENTATION

Programs written in a high-level language have rich structure. Researchers have proposed graph-based representations to capture this structure (Allamanis et al., 2018). We start with this approach of representing programs using graphs with certain modifications for our task.

As shown in the left part of Figure 2, we first parse the program's source code into an abstract syntax tree (AST) form that captures the program's syntactic structure. We then connect the leaf nodes with SuccToken edges. Unlike previous approaches, we additionally add value nodes that store the actual content of the leaf nodes, with special ValueLink edges connecting them together. The purpose of introducing this additional set of nodes is to provide a name-independent strategy for code representation and modification, which we elucidate in the next section. Hereafter, we use $g_{fix}, g_{bug}$ or $g$ in general to represent either the source code or the corresponding graph structure.

After representing the program as a graph, we use a Graph Neural Network (GNN) (Scarselli et al., 2008) to map the graph into a representation in a fixed dimensional vector space. Specifically, given

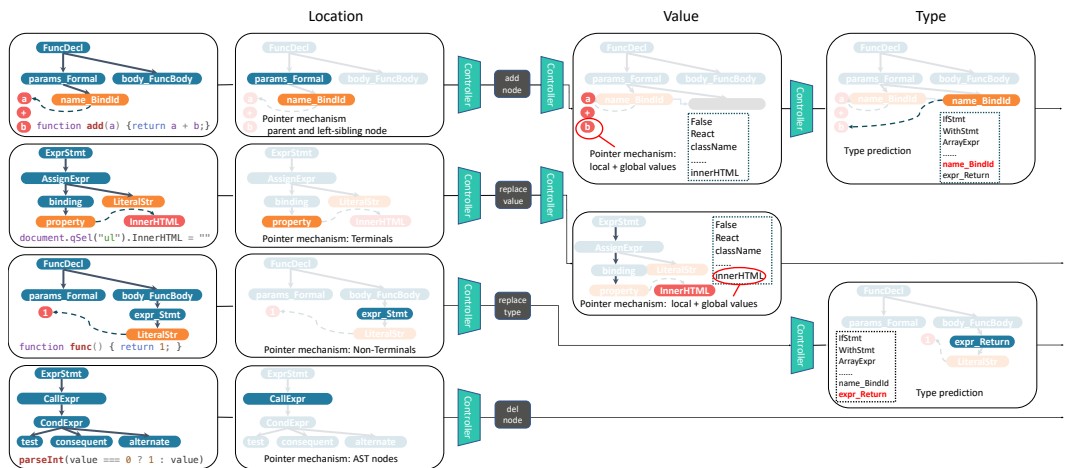

Figure 3: Graph edit operators with low-level primitives.

a graph $g = (V, E)$ with set of nodes $V$ and edges $E$, we need a function $f(g) \mapsto (\mathbb{R}^d, \mathbb{R}^{|V| \times d})$ to obtain the $d$-dimensional representation of graph $g$ (denoted as $\vec{g}$), as well as representations of individual nodes $v \in V$ (denoted as $\vec{v}$). To parameterize $f(\cdot)$, we employ the form in GIN (Xu et al., 2018), with our adaptation to our multigraph for program representation in the following manner:

$$h_v^{(l+1),k} = \sigma(\sum_{u \in \mathcal{N}^k(v)} \mathbf{W}_1^{l,k} h_u^{(l)}), \forall k \in \{1, 2, \ldots, K\}$$
$$h_v^{(l+1)} = \sigma(\mathbf{W}_2^l [h_v^{(l+1),1}, h_v^{(l+1),2}, \ldots, h_v^{(l+1),K}] + h_v^{(l)}) \tag{2}$$

where $\mathbf{W}_1^{l,k} \in \mathbb{R}^{d \times d}$, $\mathbf{W}_2^l \in \mathbb{R}^{dK \times d}$ are model parameters and $\sigma(\cdot)$ is tanh in this implementation. $K$ is the total number of edge types in this multi-graph representation. In the end, the node embedding is $\vec{v} = h_v^{(L)}$, where $L$ is the total number of propagations in the GNN. $\mathcal{N}^k(v)$ is the set of neighbors of node $v$ that are connected by edge with type $k$. Following GIN, the graph representation $\vec{g}$ is the aggregation of $h_v^l, \forall l \in 0, 1, \ldots, L$. We use max pooling to aggregate $h_v^l$ for each $l$, and then take the average of these $L + 1$ vectors to obtain $\vec{g}$.

Initially, we use the node type as one-hot features as a starting value for $h_v^{(0)}$, where the types are either obtained from the AST representation, or from the *local value table* as shown in Figure 2. Note that we don't use features like variable names or function names in this graph representation, as different programs may follow different naming conventions. Instead, we focus on the syntactic structure of the source code, so as to enable naming-agnostic representation across different programs.

## 3.2 ONE-STEP GRAPH EDIT

There are five types of operators to choose from for a single step graph edit, namely, adding a node (ADD), deleting a node (DEL), replacing a node value (REP_VAL), replacing a node type (REP_TYPE) and stop (NO_OP). When combined with multi-step edits, these operators suffice to capture a rich variety of code modifications. These operators share some common low-level primitives, such as finding the location, predicting value, etc. We first introduce the individual low-level primitives and then present how to assemble these for each type of graph edit operator.

### 3.2.1 LOW-LEVEL PRIMITIVES

Our low-level primitives contain location, type, and value prediction. These primitives can be combined for different operators later on. In this section, we assume the availability of a controller, represented as $\vec{c} \in \mathbb{R}^d$. It keeps track of the global state, including the original source code, as well as the edits made so far. We will elaborate this when we assemble different primitives together.

**Location** The *location* primitive locates a specific position in the source code. While it corresponds to region selection in the original text representation, with the graph representation, we can easily treat it as a node selection step. As different programs have different numbers of nodes, we employ a pointer network (Vinyals et al., 2015) into the graph structure. Specifically, after obtaining the node embeddings $\{\vec{v}\}_{v \in V}$, we select the node via $loc(\vec{c}, g) = \arg\max_{v \in V} \vec{v}^\top \vec{c}$ for simplicity.

**Value** The *value* primitive assigns a value for a leaf node in the AST. Instead of predicting the replacement value using a language generative model (Chen et al., 2018) or GNN score function (Allamanis

et al., 2018), we adopt the attention mechanism to let the model to choose from either the values appearing in the current file (local value table), or a collection of global values that are common for the specific language. Let $D_{val}$ be the global dictionary of commonly used leaf-node values in the language, where each item $i_v \in D_{val}$ is associated with a vector representation $\vec{i_v} \in \mathbb{R}^d$. The local value table is denoted as $V_{val}$ which is a subset of the nodes in current graph. Then, the value is predicted via $val(\vec{c}, g) = \mathrm{argmax}_{t \in D_{val} \cup V_{val}} \vec{t}^\top \vec{c}$. Again we use inner product simply for efficiency, while more expressive score functions can also be used.

**Type** The *type* primitive assigns the type for non-terminal nodes in an AST. As the total possible number of types is finite and fixed for a given language, the type prediction is simply a multi-class classification problem. However, we can utilize the AST grammar checker with contextual information to prune the output space. To predict the type of a given non-terminal node, we can obtain its parent node and current children. Then, by looping over the valid production rules at the current location, we can obtain a list of all valid types. The final type is only chosen from this set.

### 3.2.2 GRAPH EDIT OPERATORS

The $t$-th round of edit starts with the current graph $g_{t-1}$, the corresponding graph embedding, and the 'macro-context' embedding $\overrightarrow{c_M}_{t-1}$ that captures the edit history so far. Every type of edit operation (excluding `NO_OP`), requires prediction of the buggy location. So, in each round, the *location* primitive is invoked to determine the node to target. Then the edit type $e$ that is feasible at this location $v$ is predicted out of the five operators. A 'micro-context' embedding $\overrightarrow{c_{m_t}}$ is obtained from the macro embedding updated by two LSTM calls with location node embedding $\vec{v}$ and operator embedding $\vec{e}$. To summarize:

$$\overrightarrow{c_{M_t}}' = \mathrm{LSTM}(\overrightarrow{g_{t-1}}|\overrightarrow{c_{M_{t-1}}}), \overrightarrow{c_{m_t}} = \mathrm{LSTM}(\overrightarrow{e_t}|\mathrm{LSTM}(\overrightarrow{v_t}|\overrightarrow{c_{M_t}})), \tag{3}$$

The micro-context embedding is used as the controller throughout the process of each operator. In the following content, we present these operators in detail.

**ADD** This operation adds a new node to the graph. Unlike in Li et al. (2018) where the node and corresponding edges are added in separate stages, which would introduce extra complexity, we introduce a simple mechanism that can uniquely add a node and corresponding edges. As is shown in Figure 3, this process invokes one *location* primitive, one *value* primitive, and one *type* primitive. The *location* primitive invoked before the edit (*i.e.*, node $v$) determines the parent of the node to be added, while the *location* primitive called during the edit chooses the left sibling of the node. In a special case where the parent node does not have any children, then such left sibling node is set to the parent node itself. With this information, we can uniquely determine the position to insert into the AST. Finally, the corresponding edges—`SuccToken`, `ValueLink`, and AST edges—can automatically be inferred with the location of new node to be added.

As this process is autoregressive, the micro-context embedding is kept updated with all the primitive calls. For this specific operator, the context is updated in the order of: $c_{m1}^{\rightarrow} = \mathrm{LSTM}(\overrightarrow{v_{sibling}}|\vec{c_m})$, $c_{m2}^{\rightarrow} = \mathrm{LSTM}(val(\vec{c_{m1}}, g)|\vec{c_{m1}})$ and $c_{m3}^{\rightarrow} = \mathrm{LSTM}(type(\vec{c_{m2}}, g)|\vec{c_{m2}})$. In the end, $\vec{c}_{ADD} = \vec{c_{m3}}$ summarizes the process.

**DEL** This operator deletes a node and corresponding edges in the graph. If it is a non-terminal node in the AST, then the corresponding subtree is removed as well. The micro-context embedding is updated by the LSTM via the embedding of the node being deleted.

**REP_VAL** This operator replaces the value of a leaf (terminal) node in the AST. This procedure requires the prediction the value. The leaf node is linked to the new value node in the internal value table via a `ValueLink` edge. Also, the micro-context embedding is updated by the LSTM via the embedding of the corresponding node and value.

**REP_TYPE** This operator changes the type of a non-terminal node, which involves *type* primitive steps. The micro-context embedding is updated by the LSTM via the embedding of the corresponding node and type.

**NO_OP** This op does not change the graph. It simply denotes the end of the sequence of graph edits.

### 3.3 GRAPH TRANSFORMATION

Our end-to-end model for graph transformation inference is shown in Alg 1. We denote the buggy graph $g_{bug}$ as $g_0$ for simplicity. Then, for the $t$-th graph edit, the following steps are performed:

1. Obtain the graph representation $\overrightarrow{g_{t-1}}$ and node embeddings. Update the macro-context embedding using $\overrightarrow{g_{t-1}}$;
2. Choose edit location $v_t$ by performing *location* primitive and update the context embedding;
3. Pick the graph edit operator $e_t$ that is compatible with $v_t$; Use both $v_t$ and $e_t$ to obtain the micro-context embedding.
4. Perform the edit, obtain the corresponding micro-context summary $\overrightarrow{c_{e_t}}$ and update the macro-context embedding.
5. If the edit is not NO_OP, then go back to step 1; otherwise return the graph.

---

**Algorithm 1** Transformation inference of $p(g_{fix}|g_{bug})$

---

1: Input $g_{bug} \sim \mathcal{D}$ and model parameters $\theta$.
2: Obtain $\vec{g_0}, \{\vec{v}_{v \in g_{bug}}\} = f_0(g_{bug})$, let $\vec{c_{M_0}}$ be null.
3: **for** $t = 1$ to $T$ **do**
4:      Obtain $\overrightarrow{c'_{M_t}} = \text{LSTM}(\overrightarrow{g_{t-1}}|\overrightarrow{c_{M_{t-1}}})$.
5:      Choose location $v_t$, then edit type $e_t$;
6:      **if** $e_t = $ NO_OP **then**
7:          set $g_T = g_{t-1}$ and exit the loop.
8:      **end if**
9:      Perform operator $e_t$ with $\overrightarrow{c_{m_t}}$ obtained by Eq 3.
10:     Get new graph $g_t$, update $\overrightarrow{c_{M_t}}$ with $\overrightarrow{c_{e_t}}$.
11: **end for**
12: Return $g_T$

---

This process repeats until it reaches the maximum steps $T$ or the NO_OP operator is selected. Note that our framework can capture the situation when the input program is bug-free. In this case, the NO_OP operator is supposed to be triggered at the first step. Also, each edit step is not limited to a single node level operation. It can be extended to modify a certain substructure (e.g., replace a tree node with one of its children). This in turn allows program repair to be performed in fewer edit steps.

## 4 LEARNING

Given the dataset $\mathcal{D} = \{(g_{bug}^{(i)}, g_{fix}^{(i)})\}_{i=1}^{|\mathcal{D}|}$ which consists of pairs of buggy code and the fixed code, the learning objective $\max_\theta \mathbb{E}_{(g_{bug}, g_{fix}) \sim \mathcal{D}} p(g_{fix}|g_{bug}; \theta)$ maximizes the likelihood of fixes.

Since the probability is factorized according to Eq 1 where a sequence of transformations is performed, we parse the source code using the SHIFT AST format, and utilize a JSON diff toolbox to compile the code differences into a sequence of AST edits. This serves as the fine-grained supervision mechanism for our graph transformation formulation. Thus, the MLE objective above is realized with the sum of cross entropy loss at each step of graph edits. During training, we jointly optimize the graph representation module $\{f_t(\cdot)\}_{t=1}^T$, each of the operator module and the controller module which is parameterized by LSTM. We use the Adam optimizer with $\beta_1 = 0.9, \beta_2 = 0.99$ and initial learning rate of $10^{-3}$. Due to the large size of each sample, we use a small batch size of 10 during training. Furthermore, to stabilize the training, we apply the gradient clip with the maximum norm of 5.

## 5 INFERENCE

The inference procedure involves searching for the maximum in the combinatorial space: $\arg\max_{g_{fix}} p(g_{fix}|g_{bug}; \theta)$. Since the search space is very large, however, we use beam-search to approximately find the fixes with highest probabilities.

Specifically, we maintain a pool of partially fixed programs $\{\tilde{g}\}$, which starts with simply the single buggy program $g_{bug}$. The pool size is limited by the beam-search size $B$. For each $\tilde{g}$, we propose the top $B$ locations to be modified, top $B$ operators or top $B$ primitives (*location*, *type*, *value*), depending on the current stage of the edit $\tilde{g}$. Then the total joint one-step graph transformation solutions are ranked together based on the joint log-likelihood, and the top $B$ solutions with the largest likelihood are kept in the pool for the next round of beam search.

Unlike beam search for language models where the vocabulary size is fixed, in our setting, the available choices or even the steps of inference may vary (e.g., the ADD operator has more steps of primitive calls than the DEL operator). Our implementation is based on PyTorch with customized GPU kernels to enable efficient inference on GPUs.

## 6 EXPERIMENTS

**Dataset** Our model is trained and evaluated on a corpus of nearly half a million data points. We have created a robust system to continuously collect small changes in Javascript programs from Github. Given a commit, we download the Javascript file before and after the change: $(src_{buggy}, src_{fixed})$. Commits can contain many types of changes such as feature additions, refactorings, bug fixes, etc. In

|          | ADD   | REP_TYPE | REP_VAL | DEL    | total   |
|----------|-------|----------|---------|--------|---------|
| train    | 6,473 | 1,864    | 251,097 | 31,281 | 290,715 |
| validate | 790   | 245      | 31,357  | 3,957  | 36,349  |
| test     | 796   | 233      | 31,387  | 3,945  | 36,361  |

Table 1: Statistic of `OneDiff` dataset. See appendix for more information of other dataset.

|          | Total | | *Location* | | *Operator* | *Value* | | *Type* | |
|----------|-------|-------|-------|-------|-------|-------|-------|-------|-------|
|          | Top-3 | Top-1 | Top-3 | Top-1 | Top-1 | Top-3 | Top-1 | Top-3 | Top-1 |
| **TOTAL** | **26.1** | 14.2 | 35.5 | 20.4 | 34.4 | 52.3 | 29.1 | 76.1 | 66.7 |
| ADD      | 52.9 | 39.2 | 69.6 | 51.4 | 70.6 | 65.7 | 55.1 | 76.8 | 68.5 |
| REP_VAL  | 23.4 | 11.9 | 33.3 | 18.5 | 31.7 | 53.0 | 28.8 | - | - |
| REP_TYPE | 71.7 | 52.4 | 73.0 | 52.8 | 79.4 | - | - | 74.7 | 61.0 |
| DEL      | 39.6 | 24.8 | 44.0 | 27.5 | 45.8 | - | - | - | - |
| Random   | .08 | .07 | 2.28 | 1.4 | 27.7 | .01 | .01 | .27 | 0 |

Table 2: Evaluation of model on the `OneDiff` dataset: accuracy (%).

an attempt to filter our dataset to only include bug fixes, we use a heuristic based on the number of changes to the AST. Our insight is that a commit with a smaller number of AST differences is more likely to be a bug fix than a commit containing large changes. Thus for the experiments, we use three different datasets: `OneDiff` with precisely one edit; `ZeroOneDiff` with zero and one edit and `ZeroOneTwoDiff` with zero, one or two edits. We additionally filter out data points with ASTs larger than 500 nodes as a parameter in our system. A detailed overview of our corpus crawler is available in Appendix B.

### 6.1 EVALUATION

We train the model for 3 epochs on the training set until the validation loss converges. We tried different configurations of our model with different number of layers and different graph embedding methods besides the generic one in Eq 2. We report on these ablation studies in Appendix C.

Table 2 shows the evaluation results of our model on a held out test set consisting of samples from our `OneDiff` dataset. Additional experiments on `ZeroOneDiff` and `ZeroOneTwoDiff` datasets are available in Appendix A. We also provide experimental results with respect to different configurations.

Accuracy is shown for each graph edit operation type. Accuracy is measured in a complete discrete graph edit operation step. For example consider Figure 1a, in which we edit an object property name with the `REP_VAL` operation. If the model incorrectly predicts the edit operation to be of type `DEL`, then it will not go on to predict a *value*. In this case, the model will be penalized twice in the operation accuracy as well as the value accuracy. A prediction is considered totally correct only if the entire sequence of graph edit primitives is correct. Note that top-1 greedy prediction is not always among top-3 when beam search is used. Additionally, operator prediction is only evaluated on the top prediction as the search space only includes four operators.

To demonstrate the magnitude of the search space, we compare HOPPITY to a model that selects uniformly at random, in each step of the graph edit process. The random model performs well at operation type selection since the search space only has four options (`ADD`, `REP_VAL`, `REP_TYPE`, `DEL`). However, after the operation type is predicted, the random model's accuracy drops, as there are up to 500 nodes in the buggy AST. When it predicts value, the accuracy drops even further as our vocabulary contains 5,000 values. Lastly, type prediction has slightly better accuracy than value prediction because the number of the types of AST nodes in total is smaller than our vocabulary.

### 6.2 BASELINES

As existing approaches cannot be applied for comparison in Table 7, we adapt the baselines to some restricted settings in this section. We report the results on the `OneDiff` dataset as most of the baselines target repair of a single bug. Note that for all comparisons we provide equal amounts of information to HOPPITY and the baseline without retraining our model.

| Type | GGNN-Rep | GGNN-Cls | HOPPITY |
|---|---|---|---|
| Top-1 | 53.2% | **99.6%** | 90.0% |
| Top-3 | 85.8% | **99.6%** | 94.8% |

Table 3: `REP_TYPE` accuracies with location+op.

| Value | GGNN-Rep | GGNN-RNN | HOPPITY |
|---|---|---|---|
| Top-1 | 63.8% | 60.3% | **69.1%** |
| Top-3 | 67.6% | 63.6% | **73.4%** |

Table 4: `REP_VAL` accuracies with location+op.

| | Top-1 | Top-3 |
|---|---|---|
| HOPPITY | **67.7%** | **73.3%** |
| SequenceR | 64.2% | 68.6% |

Table 5: Overall OneDiff accuracy with location.

| Bug Type | Amount | TAJS | HOPPITY |
|---|---|---|---|
| Undefined Property | 7 | 0 | 1 |
| Functional Bug | 11 | 0 | 3 |
| Refactoring | 12 | 0 | 1 |
| Total | 30 | 0 | 5 |

Table 6: Comparison with TAJS.

**GGNN:** Allamanis et al. (2018), uses Gated Graph Neural Networks (GGNN) for two specific bug repair tasks: VARMISUSE, in which the model learns to select the correct variable that should be used at a given location, and VARNAMING, in which the model predicts a variable name based on its usage. We adapt these tasks to compare with HOPPITY on the `REP_TYPE` and `REP_VAL` tasks. Specifically, for `REP_TYPE` prediction we have

- GGNN-Rep: we adopt VARMISUSE to replace with candidate node type and modify the graph structure correspondingly; we use their proposed max-margin formulation for training.
- GGNN-Cls: we perform multi-class classification using the target node and graph embedding.

For `REP_VAL` prediction, we also made two versions of adaptations:

- GGNN-Rep: similar to above, here the candidate set is from values in the current graphs plus the top-100 frequent values used for repair in the training set.
- GGNN-RNN: we adopt VARNAMING approach to predict value directly. Due to the huge vocabulary size, we use char-level language model for predicting the replacement.

Table 3 and 4 show the comparison when buggy node is known. Regarding the type prediction, as the number of types is large, the likelihood formulation with classification objective outperforms the max-margin loss based one (*i.e.*, GGNN-Rep). As in this limited case GGNN-Cls and HOPPITY are quite similar except for graph representation, the performance is expected to be comparable. As HOPPITY is not trained to predict type fix only, it performs slighly worse than GGNN-Cls. Also for the value prediction, our formulation of pointer on graph is more effective. We found when the space of decisions is large, it is hard to apply structured prediction method like GGNN-Rep in this setting. Since real-world programs are noisy, the sentences used in different programs vary greatly, making it difficult for language models to predict the exact accurate value. A possible extension is to combine the language model with the graph pointer, which we will explore in future work.

**SequenceR:** The model proposed by Chen et al. (2018) is a translation based model that predicts a fixed sequence of tokens when given a buggy line in the source code. We compare with our model by providing location information to both approaches.

Table 5 summarizes the total accuracy for fixing a single bug. In order to provide a fair comparison, we allow SequenceR to predict the same information as our model (*i.e.*, predict op, value *etc.*in a sequential way), rather than an entire sequence of raw textual tokens. This experiment shows the benefit of formulating code repair with graphs over text tokens.

With the above two baselines, we can see that in the restricted case our model can still yield comparable or even better performances. Given that our model can go for more edits without location information, we believe this tool is more generic and effective for code repair.

**TAJS:** We also compare the bug detection ability of HOPPITY against TAJS (Jensen et al., 2009) which is a well-known static analysis tool for Javascript programs. Automating the comparison for our entire test set proved to be infeasible. For example, TAJS only accepts JavaScript ES5 programs, while the vast majority of current JavaScript projects use ES6 or other variants like React JSX. Another problem is that TAJS does not analyze code that is not invoked, e.g., a library function that is not called by client code. Moreover, determining the right command-line options of TAJS is non-trivial since it provides many options targeting different JavaScript runtime environments. Due to these issues, we forgo a large-scale comparison, and instead pick 30 random points in our test set to manually analyze using TAJS. Table 6 depicts the results (Appendix D provides further details).

We restrict the chosen test points to satisfy a necessary condition for undefined property bugs since TAJS claims to be proficient in detecting this class of bugs. In the process, we also pick some functional bugs, as well as cases of refactoring modifications. By resolving the numerous issues that prevented us from automating the comparison, we were able to run TAJS manually. TAJS failed to detect any real bugs in the 30 test points. While functional bugs and refactoring modifications are beyond TAJS, however, TAJS also raises many unrelated false alarms due to its failures in locating NodeJS libraries, importing JSON files, or recognizing built-in global variables. These warnings are detrimental because TAJS suspends the analysis as soon as it detects what it preceives to be a bug. To further aid TAJS, we omitted parts of each program that are unrelated to the bug, in the hope of driving TAJS's analysis as deep as possible. After all these measures, TAJS managed to detect two of the undefined property bugs (Bug IDs 4 and 6 in Appendix D).

In contrast, HOPPITY is able to correctly detect 5 bug locations of the 30 testing points within our top 3 predictions. Moreover, HOPPITY also produces 4 patches that are identical to the developer's fixes. Our comparison highlights HOPPITY's two important strengths compared to TAJS. First, HOPPITY relieves developers from the enormous burden of manual configuration. Second, HOPPITY achieves far better performance in detecting as well as fixing the bugs in Javascript programs.

## 7 RELATED WORK

**Static analysis for bug detection.** Static analyzers such as FindBugs, Error-Prone, and Semmle use syntactic pattern-matching and dataflow analysis to find common bugs. Typically, detecting even a single class of bugs can require dozens or even hundreds of patterns. Coverity (Bessey et al., 2010), SonarQube, and Clang Static Analyzer check for semantic inconsistencies in code based on more sophisticated path analyses. Infer (Calcagno et al., 2015) is built upon sound principles and can prove the absence of certain classes of bugs. TAJS belongs to this category as well. Due to the undecidability of the problem, however, approximations are inevitable which voids the guarantees in practice. Compared to all static analysis tools, HOPPITY offers the following advantages: (1) it targets a board range of programming errors; (2) it not only localizes bugs but also fixes them; and (3) it has significantly higher signal-to-noise ratio (i.e., detects more bugs with less false alarms).

**Learning-based bug detection.** Allamanis et al. (2018) target variable-misuse errors and present a solution based on a gated graph neural network model to predict the correct variable name given a buggy location. Vasic et al. (2019) present a pointer network on top of a RNN which outperforms Allamanis et al. (2018) on the same task. DeepBugs (Pradel & Sen, 2018) proposes a name-based bug detection scheme. Their model is trained to predict three classes of bugs: swapped function arguments, wrong binary operator, and wrong operand in a binary operation. Compared to these models, our approach is capable of detecting and fixing a wide range of errors in Javascript. SequenceR (Chen et al., 2018) uses sequence-to-sequence model to translate a buggy code segment into correct one; Getafix (Scott et al., 2019) produces human-like bug fixes by learning from past fixes. It employs a hierarchical clustering algorithm that sorts fix patterns according to their generality. While these approaches are general against different types of bugs, they still need the bug location as input.

**Graph learning and optimization.** Our work is closely related to the literature in graph representation learning and optimization. Our model uses a variant of GNN that is inspired by many representative works (Li et al., 2015; Xu et al., 2018; Si et al., 2018), with the adaptation of local value table and pointer mechanism. Our work is also related to auto-regressive graph modeling Johnson (2016); Li et al. (2018); Brockschmidt et al. (2018); Dai et al. (2018), but with more generic operations such as subtree deletion and attribute modifications. Some other works model the graph modification in latent space (Jin et al., 2018; Yin et al., 2018), but such frameworks lack fine-grained control over the generative process, and thus are not very suitable for performing code repair.

## 8 CONCLUSION

We proposed an end-to-end learning-based approach to detect and fix bugs in Javascript programs. We realized the approach in a tool HOPPITY and demonstrated that it correctly predicts 9,490 out of 36,361 code changes in real programs on Github. In the future, we plan to expand the targeted bugs to include those that are caused by the interdependence among multiple files or that require multiple steps to fix. We will also deploy HOPPITY in an IDE to further evaluate its accuracy and utility. Finally, we plan to extend our learning framework to support other languages. Due to its language-independence, we believe HOPPITY will benefit developers beyond Javascript as well.

## ACKNOWLEDGMENTS

We thank the reviewers for their insightful comments. This research was supported in part by NSF awards #1836936 and #1836822, ONR award #N00014-18-1-2021, and Facebook research awards.

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

# A    ADDITIONAL EXPERIMENTS

|  | Total | | *Location* | | *Operator* | *Value* | | *Type* | |
|---|---|---|---|---|---|---|---|---|---|
|  | Top-3 | Top-1 | Top-3 | Top-1 | Top-1 | Top-3 | Top-1 | Top-3 | Top-1 |
| `ZeroOneTwoDiff` | 40.8 | 29.7 | 18.9 | 3.9 | 30.3 | 35.0 | 6.5 | 38.6 | 3.4 |
| `ZeroOneDiff` | 51.6 | 34.5 | 27.1 | 5.5 | 35.6 | 45.4 | 10.4 | 73.9 | 58.9 |
| `OneDiff` | **26.1** | 14.2 | 35.5 | 20.4 | 34.4 | 52.3 | 29.1 | 76.1 | 66.7 |
| Random | .08 | .07 | 2.28 | 1.4 | 27.7 | .01 | .01 | .27 | 0 |

Table 7: Evaluation of models on each dataset. The Random model is evaluated on the `OneDiff` dataset and is shown for comparison.

**Full experiment results**    In addition to the evaluation of samples with one edit Table 7, we also evaluate HOPPITY on the following datasets:

- `ZeroOneDiff` - Includes samples with labels of zero or one edit
- `ZeroOneTwoDiff` - Includes samples with labels of zero, one, or two edits.

We trained models on each dataset for roughly 12 hours on a single GTX 2080Ti GPU. Accuracy on the `ZeroOneDiff` is the highest as predicting that an AST is not buggy does not consist of any low level primitive predictions. This makes it a much easier prediction for the model than say, an `ADD` operation which the parent location, left sibling, value, and type must all be predicted correctly in order to be considered accurate.

|  |  | TRUE LABEL | |
|---|---|---|---|
|  |  | BUGGY | NOT BUGGY |
| PREDICTED | ALARM | 10,293 | 7,210 |
|  | NO ALARM | 26,517 | 20,605 |

Table 8: Results on true/false predictions.

**False positive/negative study**    An evaluation of false positives and false negatives is available in Table  8. In this setting, we treat the problem as a classification problem on our `ZeroOneDiff` dataset and our model attempts to predict if a given AST is BUGGY / NOT BUGGY. If the model predicts `ADD`, `REP_VAL`, `REP_TYPE`, or `DEL`, we consider this a prediction of "BUGGY." Accordingly, if the model predicts `NO_OP`, we consider this to be a prediction of "NOT BUGGY."

**Accuracy v.s. size of graph**    To demonstrate the affect of AST size on HOPPITY's prediction accuracy on the `OneDiff` dataset, we include Figure 4. As expected, AST size and accuracy are inversely related.

| Beam Size (k) | Top-k Accuracy (%) |
|---|---|
| 1 | 14.37% |
| 2 | 21.10% |
| 3 | 26.14% |
| 4 | 30.12% |
| 5 | 33.58% |

Table 9: Accuracy vs beam sizes.

**Accuracy v.s. beam search size**    In Table 9 we compare the performance with different beam sizes on the `OneDiff` dataset. As we can see, the top-3 accuracy with beam size 3 is significantly better than top-1 accuracy with just greedy prediction. This is expected, as in the decision process there are 'bottleneck' stages with only a few predictions (e.g., the op prediction). Thus from beam-1 to beam-3 there's huge improvement, but further beyond the performance maxed out.

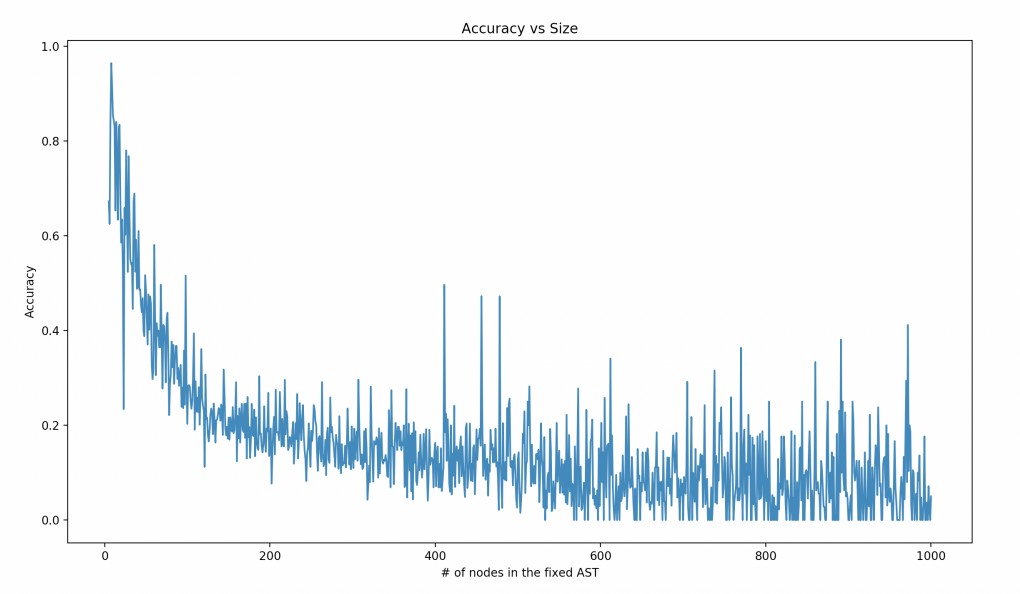

Figure 4: End-to-end code repair accuracy v.s. size of AST of source code.

## B    DATA COLLECTION

We have built a robust system to automatically collect millions of bug-fixes in Javascript programs from Github. Our system continuously crawls Github for commits containing Javascript files and creates a label consisting of the change to the AST corresponding to each such file.

Our system consists of three entirely automated parallel steps:

1. **Collect Commits:** Our system uses the GH Archive API to easily access Github event data for a specific hour in time. After obtaining all data for the hour, we filter this using the Github API to only include commits that consist of edits to Javascript files.

2. **Download Files:** As we are obtaining a list of valid commits from step 1, we begin downloading the pair: $(src_{buggy}, src_{fixed})$ where $src_{buggy}$ is the file prior to the commit, and $src_{fixed}$ is the file following the commit that contains the changes made.

3. **Create Label:** For each Javascript file downloaded, we parse the source code into a JSON format of the AST. Our system uses the SHIFT AST [1]. Abstract Syntax Tree representations are designed to naturally and intuitively represent the structure of the source code. Because of this design goal, small changes in the source code can often lead to very large changes in the AST. We chose the SHIFT AST representation with consideration to our goal of maximizing the number of commits with only one difference between the ASTs. This component produces a pair of ASTs: $(AST_{buggy}, AST_{fixed})$, at which point a JSON differencing algorithm, fast-json-patch [2] is applied to create a label. The label includes the operation type and node edited for each difference between $AST_{buggy}$ and $AST_{fixed}$.

Each step of this process is parallelized in order to grow our corpus as quickly as possible. Our dataset has the advantage that it is continuously growing without human input.

Our system is language independent and highly extensible and modular. For example, it can handle any language so long as it can be parsed into a JSON AST.

For each label, we must download two files $src_{buggy}$ and $src_{fixed}$. Additionally, if source files cannot be parsed into a SHIFT AST, a label cannot be created. For our learning corpus, we limit the dataset to only include labels with one AST difference. Additionally, in an attempt to limit graph size, we only include data points in which the $AST_{buggy}$ and $AST_{fixed}$ have less than 500 nodes.

---

[1]https://shift-ast.org/
[2]https://www.npmjs.com/package/fast-json-patch

| | |
|---|---|
| Total Files Downloaded: | 52,719,402 |
| Total Labelled Data Points: | 15,225,347 |
| # AST differences: | # data points: |
| 0 | 3,473,391 |
| 1 | 1,863,193 |
| 2-10 | 3,247,437 |
| 11-20 | 2,117,977 |
| 21-50 | 2,047,998 |
| 51-100 | 858,981 |
| 101+ | 921,754 |

Table 10: Data collection statistics.

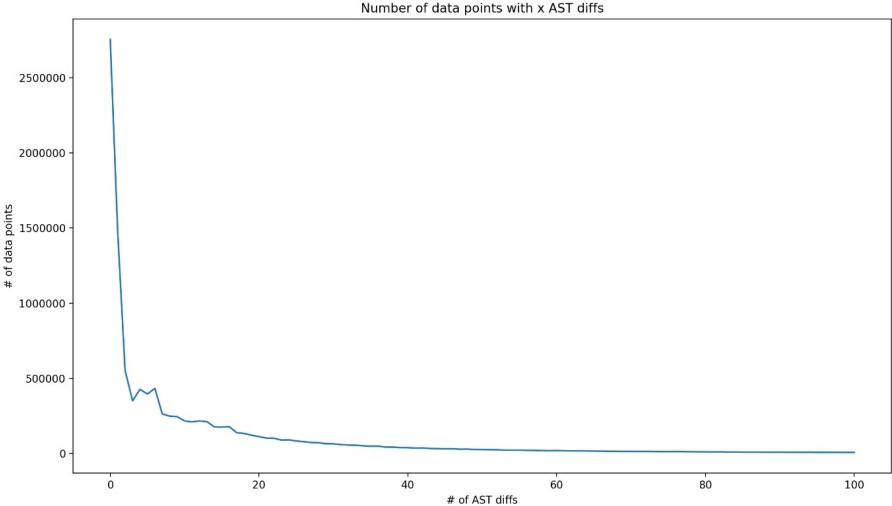

Figure 5: Distribution of number of edits in the entire crawled dataset.

Figure 5 plots the distribution of number of edits that are recorded in Table 10. We can see the distribution is long tail, with majority of edits as 1 or 2.

## C    ABLATION STUDY

We tried different graph representations with corresponding graph embedding methods. The *multi* represents the multi-graph defined by different edge types, with the parameterization of message passing function mentioned in Eq 2; the *code2inv* is the parameterization used in Si et al. (2018); the *single* instead uses a single graph with edge types as one-hot edge features. We found that more layers does not lead to better generalization in our setting, and it becomes slower in terms of convergence. So we report the results with 4 layers in our main paper.

| model | max_lv | Total | *Operator* | *Location* | *Value* | *Type* |
|-------|--------|-------|----------|----------|-------|------|
| multi | 20 | 7.63 | 30.0 | 13.1 | 22.6 | 54.5 |
| multi | 14 | 11.05 | 48.0 | 17.9 | 38.6 | 61.6 |
| **multi** | **4** | **13.33** | 53.4 | 36.2 | 38.6 | 56.4 |
| code2inv | 20 | 10.3 | 18.1 | 25.7 | 38.8 | 57.7 |
| code2inv | 14 | 8.92 | 40.0 | 18.1 | 36.0 | 55.9 |
| code2inv | 4 | 13.29 | 30.8 | 18.9 | 28.2 | 68.21 |
| single | 20 | 5.00 | 20.2 | 10.3 | 14.2 | 44.8 |
| single | 14 | 10.69 | 67.7 | 18.6 | 49.6 | 38.7 |
| single | 4 | 12.88 | 55.8 | 20.8 | 43.2 | 55.8 |

Table 11: Ablation study with different graph embedding parameterizations and different number of layers. Full end-to-end repair accuracy as well as the accuracies for each primitives are reported. All the numbers are for top-1 prediction.

# D 30 RANDOM TESTING POINTS FOR TAJS BASELINE STUDY

| ID | Github Link to Diff | File | Buggy Code | Fixed Code |
|---|---|---|---|---|
| 1 | js-ajax-hitting-apis-lab-v-000 | index.js | login | name |
| 2 | roma2hira | convert.js | (elem).value | (elem).innerHTML |
| 3 | LetsRoll | router.js | username | userName |
| 4 | MEAN | articles.server.route.js | app.params | app.param |
| 5 | js-dom-and-events-acting-on-events-lab-v-000 | index.js | (elem).InnerHTML | (elem).innerHTML |
| 6 | m-mitrais-mb | ListAlbums.js | info.type | info.album_type |
| 7 | React-QuizComponent | QuizQuestion.js | (obj).instruction_texts | (obj).instruction_text |
| 8 | musketeer-shop | order.js | DECIMAL | INTEGER |
| 9 | ALFACharts | crosshairs.js | Math.floor | Math.round |
| 10 | hackcincinnati/site | Advisors.js | color.primary | color.accent |
| 11 | react-native-with-redux-react-navigation-v2-boilerplate | splash.js | COLOR.DARK | COLOR.PANTOME |
| 12 | AloChat | Container.js | PropTypes.string | PropTypes.object |
| 13 | pandora-validation | index.js | addAsyncSetup | addSyncSetup |
| 14 | Orca | z.js | erase() | explode() |
| 15 | iotdb-mongodb | count.js | then | make |
| 16 | flom-react | question.js | DataTypes.STRING | DataTypes.TEXT |
| 17 | j5-leds | blink.js | blink | strobe |
| 18 | Craxi | display.js | game.draw() | game.run() |
| 19 | graph-js | point.js | data | dataManager |
| 20 | matic.js | getETHFromFaucet.js | ETHFaucetAddress | ETHFAUCET_ADDRESS |
| 21 | zulip | stream_muting.js | add_messages | add_old_messages |
| 22 | WTF-Adventure | mana.js | this._super | this.super |
| 23 | simple-crafting | gather.js | getBehavior | getMeta |
| 24 | geekTalks | index.js | checkTalkOwnership | checkCommentOwnership |
| 25 | pizza-totally-rocks | Form.js | getRecipe | getRecipes |
| 26 | koot | before_router_match.js | origin | originTrue |
| 27 | fo_rest | index.js | testRecipes | preference |
| 28 | cryptii | ROT13.js | registerSetting() | addSetting() |
| 29 | DiscordWithDatabase | help.js | print() | printSplit() |
| 30 | org.civicrm.civicase | CaseDetailsFileTab.js | areAvailable() | isAllowed() |

| ID | Bug Type | TAJS | HOPPITY | | | |
|----|----------|------|---------|---|---|---|
| | | False Alarm | Prediction | Top 1 | Top 2 | Top 3 |
| 1 | undefined property | Failed `doc.getElem()` | | | | |
| 2 | undefined property | | | | | |
| 3 | undefined property | Failed importing library `util` | | | | |
| 4 | undefined property | Undefined `process` | | | | |
| 5 | undefined property | Failed `doc.getElem()` | `app.params('...', ...)` | | | ✓ |
| 6 | undefined property | | | | | |
| 7 | undefined property | Undefined `process` | | | | |
| 8 | functional bug | | | | | |
| 9 | functional bug | | | | | |
| 10 | functional bug | | | | | |
| 11 | functional bug | | | | | |
| 12 | functional bug | | | | | |
| 13 | functional bug | | | | | |
| 14 | functional bug | | `serviceProvider.addAsyncSetup(....)` | | ✓ | |
| 15 | functional bug | Failed importing library `events` | | | | |
| 16 | functional bug | | | | | |
| 17 | functional bug | | `led.strobe(750)` | ✓ | | |
| 18 | functional bug | | `this.game.draw()` | ✓ | | |
| 19 | refactoring | | | | | |
| 20 | refactoring | | | | | |
| 21 | refactoring | | | | | |
| 22 | refactoring | | | | | |
| 23 | refactoring | | | | | |
| 24 | refactoring | | Failed importing library | | | |
| 25 | refactoring | | Failed importing library | | | |
| 26 | refactoring | | | | | |
| 27 | refactoring | | `exports.testRecipes = ...` | | ✓ | |
| 28 | refactoring | | | | | |
| 29 | refactoring | | | | | |
| 30 | refactoring | | Failed importing library | | | |

