# OpenReview forum: "HOPPITY: LEARNING GRAPH TRANSFORMATIONS TO DETECT AND FIX BUGS IN PROGRAMS"
_ICLR.cc/2020/Conference — Accept (Spotlight)_

### Official Review · AnonReviewer3 · 2019-10-09
**Official Blind Review #3**

**Rating:** 6

**Review:**

This paper proposes a graph tranformation-based code repair tool. By representing source code as a graph a network is asked to take a series of simple graph edit operations to edit the code. The authors show that their method better predicts edits from existing code.

Overall, I find the problem interesting and the neural approach of the authors reasonable, principled and interesting. However, the evaluation is either badly written or the authors have not understood the related literature they are comparing to. It is thus unclear how this model compares to alternatives. I believe that this is good work that needs to eventually be published, but it's not ready at this time.

* In 3.2.1 "Value" paragraph "Instead of predicting the replacement value using a language generative model (Chen et al 2018, Allamanis et al 2018), we let the model to choose from either the values appearing in the current file."

   -The Chen et al. (2018) paper is indeed a translation-style model, but thanks to the copying mechanism, the model can also choose from values appearing in the current input.
   - The Allamanis et al. (2018) is *not* a generative model and it can only select a misused variable for those that exist on within the scope (for the Variable Misuse Task)

* The authors compare to the VARNAMING (Section 6.2) task of Allamanis et al. (2018). This is odd, since to my understanding, this task is about alpha-renaming, i.e. semantics preserving renaming of all identifiers of a variable. This is not comparable with the REP_VAL action which simply replaces the value of a single leaf node from the value table. The VARMISUSE objective (picking one variable directly from the value table) is the closest analogue, although it performs no localization.

Thus, it is not surprising that a char-level language model cannot predict the names of the elements in the value table. So, I don't think that we learn anything from the current comparison.

* It's unclear to me why the REP_TYPE action is relevant to VARMISUSE task of Allamanis et al. (2018). This seems to simply be a classification task (among all known/valid types of the tree) given an existing position.

* Given the above two points, it's unclear what the comparison with the baseline (Table 3) means and what one can deduce from the evaluation: The Allamanis et al. (2018) work tries to find the correct variable at a given location; this work tries to pick a location and find a different value from current one. The Allamanis et al. (2018) and Cvitkovic et al. (2018) uses additional semantic information (as edges), importantly data flow and control flow. This work uses none of those. I am not sure that I can see how the comparison here can work.

* In 3.1 "Unlike previous approaches, we additionally add value nodes that store the actual content of the leaf nodes [...]. The purpose of introducing this additional set of nodes is to provide a name-independent strategy for code representation and modification". I am not sure what's unlike previous approaches. For example, the Allamanis et al. (2018) and the Cvitkovic et al. (2018) work, connect the same variables with various ways (e.g. data-flow edges) in a way that provides a name-independent strategy for code representation. I would fing it surprising that a principled comparison between Allamanis et al. (2018)/ Cvitkovic et al. (2018) and this work will yield any improvements: all papers use the same model (variations of a graph neural network) and this work uses strictly fewer information (no data flow, control flow edges). A comparison would simply compare how the two graph-extraction methods compare, but nothing valuable otherwise.

* The authors of this work frequently refer to Chen et al. (2018), but never compare with it. I would expect that this model beats the Chen et al. (2018) work, but this needs to be shown. In my opinion this is a much more relevant baseline, compared to the comparison with Allamanis et al. (2018). If the authors insist on comparing with that work, then a "fair" comparison needs to be made. Comparing between REP_VAL and VARMISUSE given the location of the node-to-be-repaired, is probably the only reasonable option. Comparing with Vasic et al. (2019) for the REP_VAL/REP_TYPE would also be reasonable.

* The correctness of the NO_OP predictions are never evaluated. Does the model "know" when to terminate or does it keep editing? It's unclear what is the overall accuracy (for a full edit).

Other questions:

* The "ADD" operation predicts two locations: one for the parent and one for the left sibling of the node. Does this mean that for a given node the "ADD" operation will never be able to add a left-most child (since it has no left sibling)?

* It has been recently found that code corpora collected by scraping GitHub may contain a disproportionate amount of duplicates (Lopes et al. 2017, Allamanis 2018). It is unclear if the authors have taken any steps to detect and remove duplicates that would affect their results.

* In Table 2: Does "Beam-3" mean "accuracy in the top 3" or does it mean "accuracy of the top prediction when the beam size is 3"? The difference between "Beam-1" and "Beam-3" is surprisingly large. Can you explain why?

* In the TAJS comparison, it is unclear if any negative examples are added: it's unclear what's the false positive ratio of HOPPITY vs TAJS.

Minor:

* Abstract: "Github" -> "GitHub"
* 3.2 REP_VAL: "Tthe"-> "The"
* Sec 7: "Vasic et al. (2019) present a pointer network..." It is unclear if this work outperforms the Allamanis et al. (2018) work as the comparison is only partial. Another ICLR submission https://openreview.net/forum?id=B1lnbRNtwr suggests that this is not the case.
* The citation of Chen et al. (2018) should be capitalized correctly "Sequencer"->"SequenceR"


## References

Allamanis, Miltiadis. "The Adverse Effects of Code Duplication in Machine Learning Models of Code." arXiv preprint arXiv:1812.06469 (2018).

Cvitkovic, Milan, Badal Singh, and Anima Anandkumar. "Open Vocabulary Learning on Source Code with a Graph-Structured Cache." arXiv preprint arXiv:1810.08305 (2018).

Lopes, Cristina V., et al. "DéjàVu: a map of code duplicates on GitHub." Proceedings of the ACM on Programming Languages 1.OOPSLA (2017): 84.

**Experience Assessment:**

I have published in this field for several years.

**Review Assessment: Checking Correctness Of Derivations And Theory:**

N/A

**Review Assessment: Checking Correctness Of Experiments:**

I carefully checked the experiments.

**Review Assessment: Thoroughness In Paper Reading:**

I read the paper thoroughly.

---

> ### Author Response · Authors · 2019-11-15
> **Reply to Reviewer #3 (part I)**
>
> Thank you very much for providing such comprehensive and constructive feedback! We would like to first clarify that, as there’s no direct comparison in exactly the same setting, we are trying to tweak the baselines but preserving their vanilla model at the same time. But thanks for your comments, we have adopted most of the modifications into the revision, see the details below:
>
> 1. “Value paragraph.....this is not comparable with REP_VAL…”
>
> We have rephased the statement in the paper to make it less confusing: “Instead of ... using a language generative model (Chen et al) or GNN score function (Allamanis et al), we adopt the attention mechanism to …”.
>
> Since the github data is noisy, using VARNAMING technique is straightforward but not so effective. As you suggested, we conducted additional experiments by adapting VARMISUSE technique for REP_VAL prediction. It gets 63.8% and 67.6% for top-1 and top3 accuracy, respectively. Though it is better than RNN, we found it is also hard to train with structured prediction objective (like the max-margin based one).
>
> 2. “...why the REP_TYPE action is relevant to VARMISUSE…”
>
> As changing the type of AST node would change the ordering of children in SHIFT-AST (and thus the graph), this is similar to VARMISUSE scenario. Nevertheless, we tried with multiclass classification with the concatenation of graph embedding and target node embedding. It achieves 99% accuracy which is pretty good. Our method achieves 94% which is comparable -- given that our model is trained for many more objectives.
>
> 3. “...additional semantic information…how the comparison here can work”
>
> We didn’t include control flow or SSA transform, etc, as Javascript has much less analysis support than e.g., C#. However in Table 3/4 both baseline and our method have the same level of information (ast, prev/next token, prev/next access), so the comparison still makes sense. With additional edges we would expect improvements for all the GNN based methods.
>
> We do have differences regarding the graph representation, GNN parameterization, predictive method, etc when compared with baselines. But the main purpose of this Table 3/4 is to show that, in this degenerated and special case of bug fixing, we can still get comparable performance as baseline. This serves as both sanity check and case demonstration. On the other hand, our model is a general purpose one which can go beyond this. We have refined the text to make it clear.
>
> 4. “...not sure what’s unlike…, would yield any improvements...”
>
> Regarding the representation of variable/etc, the Allamanis et al. (2018) breaks the Pascal/camel naming into subwords and feed these as bag-of-word features for nodes (see Allamanis et al. (2018),  “Initial Node Representation” section on page 5). In our paper, we didn’t use name/value content as features at all. Instead, we create additional nodes for different names/values, and use designated edge type to link AST leaves to these additional nodes.
>
> Regarding the modeling part, besides the subtle differences on GNN, we also use attention mechanism to perform REP_VAL, instead of structured prediction type of formulation. However our contribution is not mainly on these aspects, and we redo the comparison experiments with all the suggestions you made, where the results are included above and are more reasonable in this revision.
>
> 5. “...refer to Chen et al (2018), but never compare....”
>
> We agree Chen et al (2018) is more suitable, thus we have made the results available in this revision. This method needs the bug location as prior. In the end it gets 64.2% and 68.6% overall accuracy for top1 and top3 predictions. Note that we ask it to predict the same predictions as us (add/del, value, etc) in a sequential way, which is easier than predicting raw code text in their original paper.
>
> The main difficulty is the tokenizer for arbitrary (possibly broken) JS codes. We tried SHIFT-JS tokenizer but it failed. We spend lots of effort and wrote one by fixing issues with compatibility. This enables us the comparison in this revision.
>
> The limitation with Cvitkovic et al (2018) is that it replaces a single token error. We didn’t compare with this as our dataset has many more text level edits.
>
> 6. “...NO_OP predictions are never evaluated...”
>
> The dataset shown in the main paper is entirely buggy. As Table 2 reports the accuracy for one macro-step of edit, a ‘NO_OP’ would mean a code with no bug.
>
> To demonstrate the NO_OP, we have experiments in Appendix A which includes code repairs with variable number (include 0, i.e, bug free code) of edits.  As 0-edit only require predicting the code to be bug free, it is much easier than other edits with sequence of micro stages. That’s why a 0+1+2-edit dataset would get better results than purely buggy dataset 1-edit.
>
> ------------------------------
> |   dataset     |  top-3 |
> |  1-edit         |  26.1   |
> |  0+1-edit     |  48.1   |
> |  0+1+2-edit |  33.1   |
> ------------------------------

---

> > ### Author Response · Authors · 2019-11-15
> > **Reply to Reviewer #3 (part II)**
> >
> > “Other questions;”
> >
> > 1. “...ADD operation will never...left-most child”
> >
> > We enabled the left-most adding by predicting the left sibling as the parent node itself.
> >
> > 2. “...Github...duplication…”
> >
> > We have double checked the dataset and it indeed contains duplications. For example the github will repeat the edits in ‘merge’ commits, even though such commits have unique ids and dates. Thanks for pointing this out. We have rebuild the dataset by excluding the duplications, and also rerun all the experiments. The overall performance is roughly the same, since it is already hard to overfit the noisy data.
> >
> > 3. “...Beam-3...top3...difference..large..”
> >
> > Originally we use Beam-3 to refer to top-3 accuracy with beam size 3. To make it less confusing, we update the tables with top-1 and top-3, both with beam size 3.
> >
> > We think the main reason for this difference is that, the #ops is small. If top-1 predicts a wrong op, then there will be no chance to correctly predict entire fix. We further increased the beam size up to 10, and found it didn’t improve much after 3.
> >
> > 4. “...TAJS...negative examples…”
> >
> > We mainly compare TAJS in the case where all the programs are buggy, so in this setting we don’t provide negative examples. It requires more manual effort for TAJS, but we will estimate the false positive rate in our revisions.

---

### Official Review · AnonReviewer1 · 2019-10-23
**Official Blind Review #1**

**Rating:** 8

**Review:**

The paper proposes to learn from bugfixing commits to fix errors in other code. There are several good contributions of the paper, but the main one is to design a language of instructions that fix a program and to formulate the prediction to be a sequence of such instructions. The model, however, is insightful in other ways as well. The lack of naming features or generation of names makes it to point to other identifiers in a program - a major problem for most models for code. Instead, the proposed model only builds embeddings from the structure around the variables.

The work is also well evaluated, on real dataset and also it attempts to compare to the best available static analysis tools. The kind of bugs addressed by the work was also not considered in previous papers. One thing that comes to the examples of the discovered bugs is that even when the fix is shown, the user would still need detailed explanations on why was it a bug.

So, I also have questions for the authors:
1. Do you think it is possible to observe a similar reasoning to the reasoning in your text for why the buggy examples from Figure 1 are wrong, if the activations of the neural network are exposed or with other NN debugging technique.
2. It seems that a specific sequence of actions is provided in the training data and that sequence is left-to-right edits to apply the fix. In this case, doesn’t it make sense to apply restrictions on the Location primitive similar in spirit to the attention masking (see here: http://jalammar.github.io/illustrated-gpt2/#part-2-illustrated-self-attention )

Minor:
- NO_OP was used in some places (pages 4 and 5) and STOP in others (figure 2).
- N_k(v) is not defined.


**Experience Assessment:**

I have published one or two papers in this area.

**Review Assessment: Checking Correctness Of Derivations And Theory:**

I carefully checked the derivations and theory.

**Review Assessment: Checking Correctness Of Experiments:**

I carefully checked the experiments.

**Review Assessment: Thoroughness In Paper Reading:**

I read the paper thoroughly.

---

> ### Author Response · Authors · 2019-11-15
> **Reply to Review #1**
>
> Thank you for your recognition to our work and valuable suggestions! To answer your questions:
>
> 1. “...reasoning...activation of the neural network...NN debugging...”
>
> Our approach uses pointer mechanism to find the bug location, the replacement, etc. So the normalized probability given by the pointer can somehow explain the ‘reasoning’ process during the code repair. On the other hand, it is also possible to adopt attention mechanism in GNN (like graph attention network) to see how the local context contribute to each prediction. We think the explanation itself is an interesting and independent direction for a future work.
>
> However in general, we think it is hard to explain whether the prediction made by NN is indeed a good repair or not. As neural networks can sometimes be overconfident on wrong predictions (e.g., adversarial cases).
>
> 2. “...restrictions on the location primitive...”
>
> Thanks for pointing out the attention masking. In our model, the model itself is actually sequential and autoregressive. So unlike the batching + attention masking used in transformer, here doing batching wouldn’t increase the parallelism. We have actually adopted the attention masking in another aspect: for example when choosing the sibling node, we masked out all other irrelevant nodes to normalize the probability.
>
> But this question indeed raises two interesting future directions -- 1) one is the ordering of the edits. Right now we just assume a fixed order of actions. But indeed the code repair can be a latent variable to be learned together; 2) making an auto-regressive model trained in parallel is a way to speed up our training as well.
>
> “Minor:”
>
> Thanks for pointing out the issues with notations. We have fixed these in the paper.

---

### Official Review · AnonReviewer2 · 2019-10-23
**Official Blind Review #2**

**Rating:** 6

**Review:**

JavaScript is the standard programming language of the Web; according to the stats it is used on 95% of the (at least) 1.6 billion websites in the world.  Compared to other programming languages, it posses certain unique characteristics which are also responsible for making it so popular for the Web.   In this interesting work the authors aim to provide a novel data-driven system for detecting and automatically fixing bugs in Javascript.  The authors provide motivating examples behind their research. Indeed, Javascript poses unique challenges as described eloquently in Section 2, and there is a lot of space for improvement. Static analyzers have non-trivial limitations, while there is space for improving data-driven approaches. This is what Hoppity aims to achieve by translating the program into a graph, embedding the nodes and the graph as points in a d dimensional space using graph neural networks, and then using a controller that uses LSTMs decide the action to be taken among a predefined set of possible actions or a single step graph edit. These actions are reasonable, and are able to fix many bugs assuming the right sequence of actions is performed. For the purposes of learning, the author(s) use a corpus crawled and preprocessed from Github that contains more than half a million programs. Overall, I found this paper very interesting to read, and with large potential impact in practice. The paper contains a solid engineering effort, starting from the dataset collection and its preprocessing, to using state-of-the-art machinery to develop Hoppity. Therefore, I support its acceptance. However, some things were not clear from the writeup, and I hope the author(s) of the paper can give some insights.

- What is the effect of parameter T, i.e., the number of iterations? How is it set in the experiments? Clearly, the authors have a knowledge of what T should be since they have preferred programs with fewer commits.
- Following on my previous point, given the sequence of changes/graph transformations you perform, what is the distribution of the 'edit distance' (i.e., number of hops to fix a bug) in the dataset that you have? While the author(s) have
already provided a lot of stats in the Appendix, it would be interesting to see such a plot. This distribution could be insightful and serve as a rule of thumb for understanding the effect of T.
- What is the effect of the beam size? Can you plot  the accuracy  as a function of the beam size?
- Have you tried points with more than 500 nodes to see how the size of these graphs affect the performance of Hoppity?
- Can you provide further details on the running times and the GPU specs?
- The evaluation does not put enough emphasis on false positives/false negative analysis. Is it the case that bug-free programs are treated as such?
-  Have you tried Hoppity on other programming language(s)?  Do you expect such an improved performance over baselines for other languages (e.g., C++) as well?



**Experience Assessment:**

I do not know much about this area.

**Review Assessment: Checking Correctness Of Derivations And Theory:**

N/A

**Review Assessment: Checking Correctness Of Experiments:**

I carefully checked the experiments.

**Review Assessment: Thoroughness In Paper Reading:**

I read the paper thoroughly.

---

> ### Author Response · Authors · 2019-11-15
> **Reply to Review #2**
>
> Thank you for your detailed and valuable review. We address your questions and comments as follows:
>
> 1. “...effect of T...how is it set...”
>
> We trained and tested on three datasets with max T set  to 1 or 2. The T is set to be the maximum number of macro-edits needed in the training dataset. When serving the model, we simply set it to be the maximum of what it has been trained.
>
> 2. “...distribution of the edit distance...”
>
> We created a plot to visualize the statistics in Figure 5 in appendix. In general it is long tail -- our dataset has an inverse relation to number of AST differences. As we stated in Section 6, we filter our dataset to only include samples with a small number of differences. We do this due to
> 1) the availability of data. Majority of the data has 0-2 edits.
> 2) reducing noise. We adopt a heuristic that a commit with a small number of differences is more likely to be a bug than a feature addition.
> 3) the cascaded error across multiple edits.
>
> 3. “...effect of the beam size ...”
>
> We have added a table 9 in the appendix. Generally larger beam size + larger topk would give better results. As the improvement from top-1 in beam-1 to top-3 in beam-3 is significant, we think the main reason is due to some ‘bottleneck’ stage that has only a few options -- e.g., the selection of 5 operators. If top-1 gets wrong top-1 operator the entire edit would be wrong; however top-3 has more chances to make it correct. As expected, the performance didn’t go up with larger beams for top-3 prediction.
>
> 4. “...more than 500 nodes...affect...Hoppity”
>
> We have added Figure 5 in appendix to answer this question. Generally the larger the code, the harder it is to get entire edits correct, which is expected.
>
> 5. “..run time...GPU specs..”
>
> We roughly train our model for 12 hours on a single GTX 2080Ti. But the inference is fast, and we have tested it in a web-serving scenario where the latency for the entire pipeline (transmitting, parsing, constructing graph, predicting and converting back to text code) is roughly 5 seconds. So this approach is realistic regarding the efficiency.
>
> 6. “...emphasis on false positives/false negative analysis...”
> We have added the statistics below in appendix as well.
>                          BUG         NO-BUG
>                         --------------------------
> ALARM          |   8,708   |  698      |
> NO ALARM   |   14,502 |  12,459 |
>                         ---------------------------
> In general we have low false positives, but high false negatives. We think the main reason is due to the imbalance of dataset, as NO-BUG == predicting NO_OP at first hand. This would put too much weight on this op compared to other 4 ops. We think probably having an extra designated model for this binary classification would be helpful.
>
> 7. “...Hoppity on other language...”
>
> We don’t have a dataset for any other languages, so we don't think we can answer this with any experimental results. However generally our framework can support any language, as long as there exists a tool for parsing, etc. On the other hand, we choose JS based on the intuition that small edits in JS are more likely to fix semantic bugs. In a statically typed language such as Java, many of these small errors (property does not exist, incorrect operator, no return statement) would have been caught at compile time. Thus for statically typed language the need of general purpose DNN tool is less than JS.

---

### Author Response · Authors · 2019-11-15
**Summary of Revision**

Here is a brief summary of what we have changed in this paper revision. We’ve highlighted the modified content with blue color in the paper for better readability.

Dataset:
As discussed in the appendix, our dataset is continuously increasing. Also as reviewer 3 pointed out, the data generated by GhArchive has duplication. So we updated the dataset with new samples and removal of duplications, and thus updated all the relevant experiments.

Experiments:
We’ve made three splits of datasets with different # edits to study different aspects of the model.

We adapted Allamanis VARMISUSE for rep_val, and a classification formulation for rep_type as suggested by R3.

We added baseline comparison with SequenceR in location given scenario, as suggested by R3.

We’ve added more detailed studies, as suggested by R2 and R3:
- false positive/negative study, Table 8
- accuracy v.s. size of source code, Figure 4
- accuracy v.s. beam size and top-k, Table 9
- distribution of T, Figure 5

Paper:
We’ve updated the writing regarding the dataset, experiments, and also some typo fixes.

---

### Decision · Program_Chairs · 2019-12-19

**Decision:**

Accept (Spotlight)

**Comment:**

This paper presents a learning-based approach to detect and fix bugs in JavaScript programs. By modeling the bug detection and fix as a sequence of graph transformations, the proposed method achieved promising experimental results on a large JavaScript dataset crawled from GitHub.

All the reviews agree to accept the paper for its reasonable and interesting approach to solve the bug problems. The main concerns are about the experimental design, which has been addressed by the authors in the revision.

Based on the novelty and solid experiments of the proposed method, I agreed to accept the paper as other revises.